# Cytotoxic Polyketides from the Marine Sponge-Derived Fungus *Pestalotiopsis heterocornis* XWS03F09

**DOI:** 10.3390/molecules24142655

**Published:** 2019-07-22

**Authors:** Hui Lei, Jing Lei, Xuefeng Zhou, Mei Hu, Hong Niu, Can Song, Siwei Chen, Yonghong Liu, Dan Zhang

**Affiliations:** 1School of Pharmacy, Southwest Medical University, Luzhou, Sichuan 646000, China; 2CAS Key Laboratory of Tropical Marine Bio-resources and Ecology/Guangdong Key Laboratory of Marine Materia Medica, South China Sea Institute of Oceanology, Chinese Academy of Sciences, Guangzhou 510301, China

**Keywords:** marine-derived fungus, cytotoxicity, polyketide

## Abstract

Four new compounds, including two new polyketides, heterocornols M and N (**1**, **2**), and a pair of epimers, heterocornols O and P (**3**, **4**), were isolated from the fermentation broth of the marine sponge-derived fungus *Pestalotiopsis heterocornis* XWS03F09, together with three known compounds (**5**–**7**). The new chemical structures were established on the basis of a spectroscopic analysis, optical rotation, experimental and calculated electronic circular dichroism (ECD). All of the compounds (**1**–**7**) were evaluated for their cytotoxic activities, and heterocornols M-P (**1**–**4**) exhibited cytotoxicities against four human cancer cell lines with IC_50_ values of 20.4–94.2 μM.

## 1. Introduction

Marine-derived microorganisms have been attracting widespread attention as a huge resource for the discovery of bioactive natural products [1]. The genus *Pestalotiopsis*, widely distributed throughout the world, is known to produce a wide range of structurally diverse secondary metabolites with various bioactivities including polyketides [2,3,4,5], terpenoids [6,7,8], alkaloids [9,10] and others, with diverse bioactivities including antitumor, antibacterial, acetylcholinesterase inhibitory, antioxidant and anti-HIV.

Recently, we reported sixteen new compounds including heterocornols A-L [11], pestaloisocoumarins A and B, isopolisin B, and pestalotiol A [12], from the fungus *Pestalotiopsis heterocornis* (XWS03F09), which have been shown to exhibit cytotoxic and antibacterial activities in vitro. In order to obtain more new bioactive natural products from species of this genus, a strain of *Pestalotiopsis heterocornis* was refermented in the same culture medium. The chemical investigation of this fungus led to the isolation and identification of four new polyketide derivatives, heterocornols M-P (**1**–**4**), together with three known compounds, (*R*)–3–Hydroxy–1–[(*R*)–4–hydroxy–1,3–dihydroisobenzofuran–1–yl]butan–2–one (**5**) [13], (*R*)–3–Hydroxy–1–[(*S*)–4–hydroxy–1,3–dihydroisobenzofuran–1–yl]butan–2–one (**6**) [13], 4,8–dihydroxy–1–tetralone (**7**) [14] (Figure 1). The structures of the new compounds, including absolute configurations, were elucidated on the basis of spectroscopic data and a CD Cotton effects analysis. The details of the isolation, structures elucidation, and cytotoxic activity against the human cancer cell lines of these compounds are reported herein.

## 2. Results and Discussion

### 2.1. Structure Elucidation of the Compounds

Compound **1** was obtained as white amorphous powder and possessed a molecular formula of C_15_H_20_O_3_ as defined by the ^13^C-NMR and HRESIMS data (Appendix A). The ^1^H-NMR spectrum (Table 1) (Appendix A) data displayed notable signals including two methyls (δ_H_ 1.73, s; δ_H_ 1.71, s), two oxygenated methylenes (δ_H_ 5.02, dd, *J* = 12.1, 2.3 Hz, δ_H_ 4.95, d, *J* = 12.1 Hz; δ_H_ 3.73, m, δ_H_ 3.66, ddd, *J* = 11.0, 7.7, 4.7 Hz), two methylenes (δ_H_ 3.18, dd, *J* = 15.7, 7.0 Hz, δ_H_ 3.25, dd, *J* = 15.7, 7.1 Hz; δ_H_ 1.78, m, δ_H_ 2.04, ddd, *J* = 14.8, 7.7, 2.4 Hz), and an oxygenated methine (δ_H_ 5.39, m). In addition, two aromatic protons at δ_H_ 6.60 (d, *J* = 8.0 Hz) and δ_H_ 6.87 (d, *J* = 8.0 Hz) were observed. The general features of its NMR spectroscopic data closely resembled those of heterocornol I [11]. The major difference was the absence of two oxygenated methine bonds, and the presence of an oxygenated methylene (δ_H_ 3.73, m, δ_H_ 3.66, ddd, *J* = 11.0, 7.7, 4.7 Hz) in **1**. Furthermore, HMBC correlations from H-9 (δ_H_ 2.04 and 1.78) to C-8 (δ_C_ 82.7) and C-7 (δ_C_ 142.6), from H-10 (δ_H_ 3.73 and 3.66) to C-9 (δ_C_ 38.9) and C-8 (δ_C_ 82.7), and from H-11 (δ_H_ 3.25 and 3.18) to C-12 (δ_C_ 124.1), C-5 (δ_C_ 130.4), C-6 (δ_C_ 127.0), and C-7 (δ_C_ 142.6) confirmed the planar structure of **1**, as shown in Figure 1. The relative configuration of **1** was deduced by comparing with previous reports on related fungal metabolites [11,13]. A negative Cotton effect at 207 nm (Figure 2) was observed in the CD spectrum of **1**, suggesting an 8*R* configuration. Thus, compound **1** was determined, and named heterocornol M.

Compound **2** was isolated as a white amorphous solid. Its molecular formula was determined as C_14_H_18_O_4_ on the basis of HRESIMS at *m*/*z* 273.1120 [M + Na]^+^ (calcd 273.1103). A comparison of the ^1^H and ^13^C-NMR data with those of heterocornol C [11] revealed a high degree of similarity. The major difference was due to the presence of signals of –CH_2_OHCH_3_ moiety in **2**. Compared to heterocornol C, compound **2** showed an additional oxygenated quaternary carbon (δ_C_ 103.8) atom signal instead of the methine signal. The HMBC correlations from H-11(δ_H_ 3.94) to C-12 (δ_C_ 14.3) and C-10 (δ_C_ 103.8) and from H-13 (δ_H_ 3.90) to C-14 (δ_C_ 14.8), C-10 (δ_C_ 103.8), and C-9 (δ_C_ 130.1) revealed that the –CH_2_OHCH_3_ moiety was located at C-10 (δ_C_ 103.8) in **2**.

According to the literature [15,16], the absolute configuration of typical pestalospirane B was determined using NOE, a total synthesis, and an X-ray crystallographic analysis. The similar CD spectral data between compound **2** and pestalospirane B indicated that **2** (C-11 and C-13) possessed the same relative configuration as pestalospirane B (C-12, C-12′). In addition, considering the possible biosynthesis of **2**, we suggest that C-11 and C-13 are also in the *S*-configuration. Moreover, the NOESY spectrum revealed the relative configuration of **2**, as shown in Figure 3. Based on the above discussion, the absolute configuration of **2** was concluded to be 11*S*,13*S*, in that the calculated ECD curve of (11*S*,13*S*)-**2** had a good agreement with the experimental data (Figure 4). Thus, the structure of **2** was assigned, and named heterocornol N.

The same molecular formula of C_15_H_18_O_4_, as established by HRESIMS and their very similar NMR data (Table 2), suggested that compounds **3** and **4** are a pair of epimers in a nearly 1:1 ratio. The separation of compounds **3** and **4** by several chromatographic methods was unsuccessful. Fortunately, the NMR data for **3** and **4** were similar to those of **2**, with the exception of an additional six-membered ring with a hemiacetal group. In addition, the chemical shifts of C-15 (δ_C_ 39.4/37.7), C-14 (δ_C_ 97.4/95.9), and C-13 (δ_C_ 26.1/26.6) were observed. These data indicated that **3** and **4** were two hemiacetal derivatives of **2**, which was further supported by the HMBC correlations of H-15 (δ_H_ 2.32/2.42) to C-1 (δ_C_ 65.8/65.3), C-14 (δ_C_ 97.4/95.9), and C-2 (δ_C_ 123.3/120.7), and H-13 (δ_H_ 1.57/1.60) to C-15 (δ_C_ 39.4/37.7) and C-14 (δ_C_ 97.4/95.9) (Figure 5). Due to the lack of material, the configurations of C-11 and C-15 in **3** and **4** could not be determined from single crystals and the modified Mosher’s method. Thus, the structures of **3** and **4** were assigned, and named heterocornol O (**3**) and heterocornol P (**4**).

The known compounds **5**–**7** were identified as (*R*)–3–Hydroxy–1–[(*R*)–4–hydroxy–1,3–dihydroisobenzofuran–1–yl]butan–2–one (**5**) [13], (*R*)–3–Hydroxy–1–[(*S*)–4–hydroxy–1,3–dihydroisobenzofuran–1–yl]butan–2–one (**6**) [13], and 4,8–dihydroxy–1–tetralone (**7**) [14] by a comparison of the ^1^H and ^13^C-NMR, and of the MS spectroscopic data with those reported.

### 2.2. Biological Activities

All of the compounds were evaluated for their cytotoxic activities against four human cancer cell lines (Table 3). Heterocornols M-P (**1**–**4**) exhibited cytotoxicities against four human cancer cell lines with IC_50_ values of 20.4–94.2 μM. Compound **2** showed no cytotoxicity against the four human cancer cell lines when tested at 100 µM.

## 3. Materials and Methods

### 3.1. General Experimental Procedures

Optical rotations were measured with an AntonPaar MCP 200 automatic polarimeter. Ultraviolet (UV) spectra were obtained on a UV-2550 spectrophotometer (Shimadzu Corporation, Tokyo, Japan). IR spectra were recorded on a Bruker Tensor 27 FT-IR spectrometer (film). 1D and 2D-NMR spectra were carried out on a Bruker AM-400 and an Avance-500 spectrometer, δ in ppm rel. to TMS, *J* in Hz. ESIMS and HRESIMS were measured with a Bruker miXis TOF-QII mass spectrometer (Bruker, Fällanden, Switzerland), respectively. Silica gel (100–200 mesh, 300–400 mesh, Qingdao Marine Chemical Ltd., Qingdao, China), Sephadex LH-20 (GE Healthcare Bio-sciences AB, Uppsala, Sweden), and YMC GEL ODS-A (S-50 μm, 12 nm) (YMC Co., Ltd., Kyoto, Japan) were used for column chromatography. Semipreparative HPLC analyses were performed using an ODS column (YMC-ODS-A, 250 × 20 mm, 5 μm). Circular dichroism (CD) spectra were measured on a Chirascan circular dichroism spectrometer (Applied Photophysics Ltd., Leatherhead, UK). MTT assays were analyzed using a microplate reader (BioTek Synergy H1, BioTek Instruments, Inc., Vermont, USA).

### 3.2. Fungal Material

The culture of *Pestalotiopsis* sp. XWS03F09 was isolated from the sponge *Phakellia fusca*, which was collected from the Xisha Islands of China. The strain was identified as *Pestalotiopsis heterocornis* by Xiuping Lin based on DNA amplification and an ITS region sequence analysis. The result showed that the sequence was most similar (100%) to the sequence of *P. heterocornis* (GenBank database). The strain (No. XWS03F09) was deposited in the School of Pharmacy, Southwest Medical University, Luzhou, Sichuan, China.

### 3.3. Fermentation, Extraction, and Isolation

The strain *Pestalotiopsis heterocornis* XWS03F09 was inoculated into 1000 mL conical flasks containing a seed medium composed of rice at 200 g/L of, artificial sea salt at 5 g/L, and distilled water at 200 mL/L. The mass fermentation of this fungus was cultivated statically for 60 days. The solid cultures were extracted with EtOAc four times at room temperature, after which the EtOAc solutions were evaporated under reduced pressure to afford 52.2 g of crude extract.

The crude extract was subjected to silica gel column chromatography (CC) (PE-EtOAc (50:1 to 0:1, *v*/*v*) to yield 7 fractions (Frs.1–7). Fraction 4 was fractionated with repeated CC on a silica gel column eluting with a gradient of PE-EtOAc (8:1 to 0:1, *v*/*v*) to produce three subfractions (Frs. 4.1–4.3). Fr. 4.1 was chromatographed on Sephadex LH-20 eluted with MeOH to afford three subfractions (Frs. 4.1.1–4.1.3). Fr. 4.1.2 was further separated by preparative HPLC with MeOH–H_2_O (75:25, *v*/*v*) to yield **1** (10.0 mg). Fr. 4.1.3 was further separated by ODS CC, eluting with MeOH–water (80%) to yield compounds **3**/**4** (3.5 mg). Fr. 4.2 was isolated by CC on silica gel eluted with CH_2_Cl_2_–Acetone (20:1 to 0:1, *v*/*v*) to afford five subfractions (Frs. 4.2.1–4.2.5). Fr. 4.2.2 was separated by ODS CC, eluting with MeOH–water (70:30, *v*/*v*) and further purified by semipreparative HPLC (65% MeOH/H_2_O) to yield **2** and **7.** Fraction 5 was separated using silica gel column chromatography eluting with CH_2_Cl_2_–Acetone (15:1) to yield five subfractions (Frs. 5.1–5.5). Fr. 5.4 was subjected to repeated column chromatography (Sephadex LH-20) and further purified by semipreparative HPLC (60% MeOH/H_2_O) to give compounds **5** (12.0 mg) and **6** (4.5 mg).

*Heterocornol M (**1**)*: white amorphous powder; [α${]}_{D}^{25}$ + 13.3 (*c* 0.60, MeOH); UV (MeOH) λ_max_ (log ε) 232 (4.21) nm; IR (film) ν_max_ 3369, 2921, 1600, 1499, 1454, 1375, 1297, 1063 cm^−1^; ^1^H NMR and ^13^C-NMR data, see Table 1; HRESIMS *m*/*z* 271.1312 [M + Na]^+^ (calcd for C_15_H_20_O_3_Na, 271.1310).

*Heterocornol N (**2**)*: white amorphous solid; [α${]}_{D}^{25}$ − 36.0 (*c* 0.50, MeOH); UV (MeOH) λ_max_ (log ε) 257 (4.42), 305 (4.26) nm; IR (film) ν_max_ 3386, 2921, 2852, 1663, 1583, 1463, 1372, 1286, 1037 cm^−1^; ^1^H-NMR and ^13^C-NMR data, see Table 1; HRESIMS *m*/*z* 273.1120 [M + Na]^+^ (calcd for C_14_H_18_O_4_Na, 273.1103).

*Heterocornols O and P (***3**, **4***):* white amorphous solid; UV (MeOH) *λ*_max_ (log *ε*) 212 (4.55), 251 (4.33), 303 (4.22) nm; IR (film) *ν*_max_ 3359, 2921, 2852, 1659, 1632, 1580, 1457, 1412, 1377, 1339, 1173, 1098 cm^−1^; ^1^H and ^13^C-NMR data, see Table 2; HRESIMS *m*/*z* 285.1231 [M + Na]^+^(calcd for C_1__5_H_18_O_4_Na, 285.1103).

### 3.4. Calculation of ECD Spectra

A conformational analysis was performed by Spartan’s 14 using the Merk Molecular Force Field (MMFF) (Tripos, San Francisco, CA, USA) level. The low energy conformations (Boltzmann distribution ≥ 5.0%) of the compounds were submitted to the density functional theory (DFT) optimization at the level of b3lyp/6-31g(d,p), using the cpcm solvation model with the dielectric constant representing methanol in the Gaussian 09 software. The optimized structures were further submitted to the Time-dependent density functional theory (TDDFT) calculations at b3lyp/6-31g(d,p). The ORD values were read and Boltzman averaged using the SpecDis 1.53 (Gaussian, Berlin, Germany).

### 3.5. Cytotoxicity Assay

The cytotoxic activities of compounds **1**–**7** were tested in human carcinoma cell lines including a human carcinoma cell line (Ichikawa), a human gastric carcinoma cell line (BGC-823), a human liver cancer cell line (HepG2), and a human kidney cancer cell line (7860) [11]. The procedure of the cytotoxic activities evaluation described was as previously reported [11]. Adriamycin was assayed as a positive control.

## 4. Conclusions

In conclusion, seven compounds, including four new compounds, namely heterocornols M-P (**1–4**), together with three known compounds (**5**–**7**), were isolated from the marine-derived fungus *P. heterocornis*. The structures of the isolated compounds were elucidated through the detailed analysis of spectroscopic data as well as CD Cotton effects. All of the isolated compounds (**1–7**) were evaluated for their cytotoxic activities. Compounds **1**, **3**/**4**, **5**, and **6** showed cytotoxicities against four human cancer cell lines with IC_50_ values of 20.4–94.2 μM.

## Figures and Tables

**Figure 1 molecules-24-02655-f001:**
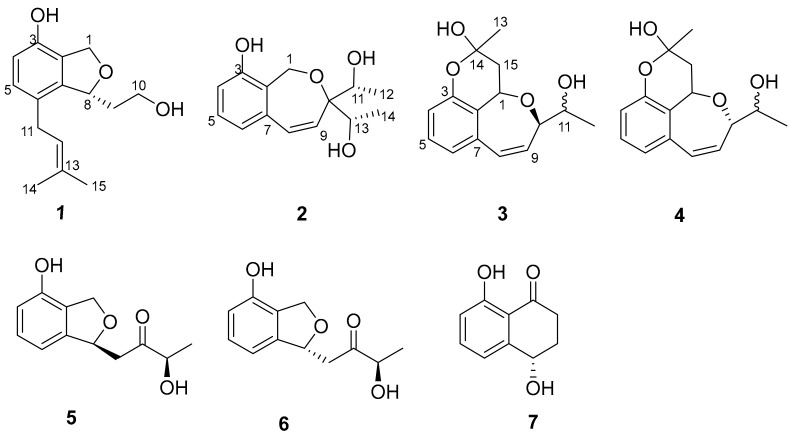
Structures of compounds **1**–**7**.

**Figure 2 molecules-24-02655-f002:**
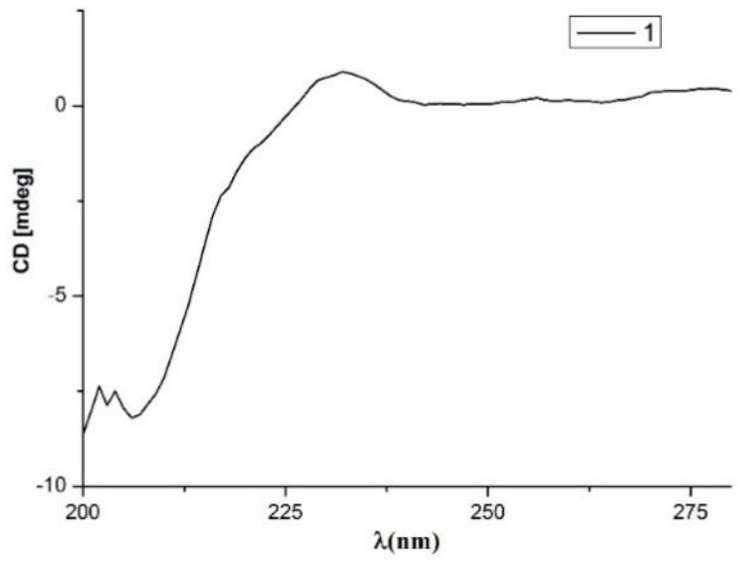
The circular dichroism (CD) spectra of **1**.

**Figure 3 molecules-24-02655-f003:**
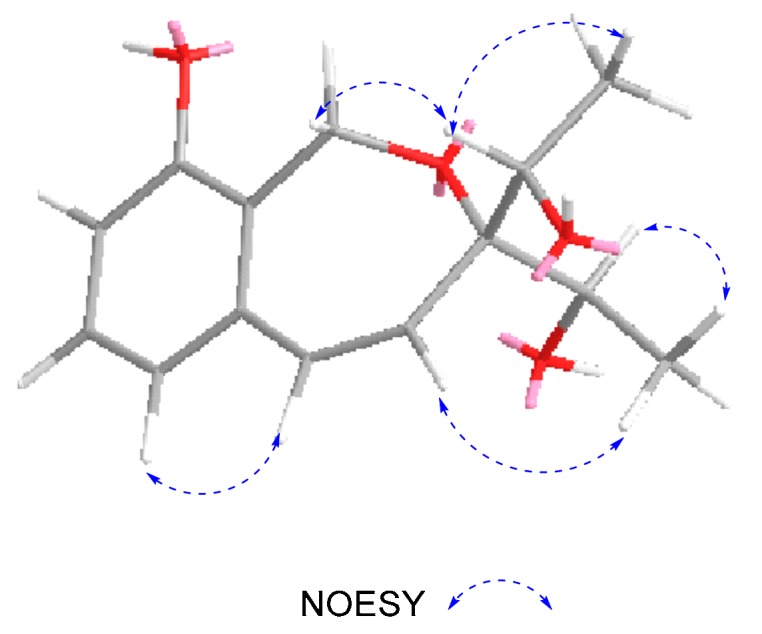
NOESY correlations of **2**.

**Figure 4 molecules-24-02655-f004:**
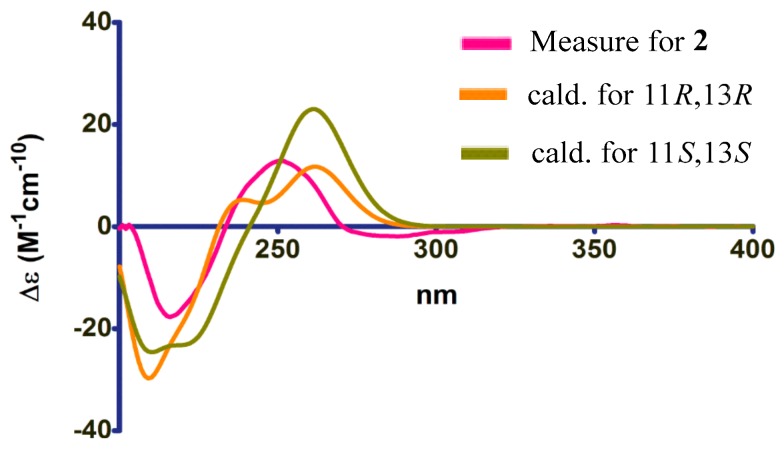
The ECD spectra of **2**.

**Figure 5 molecules-24-02655-f005:**
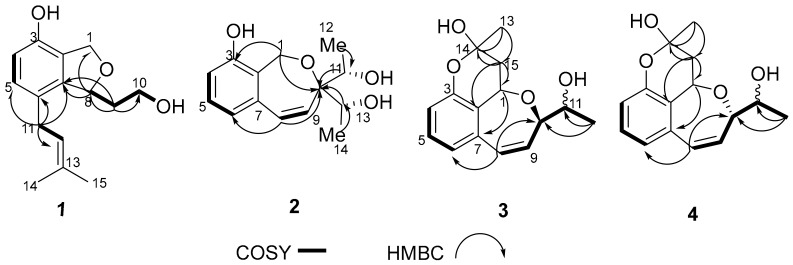
^1^H-^1^H COSY and HMBC correlations of **3**/**4**.

**Table 1 molecules-24-02655-t001:** ^1^H (500 MHz) and ^13^C-NMR (125 MHz) data for compounds **1** and **2** in CD_3_OD.

Position	1		2	
	δ_C_, Type	δ_H_ (*J* in Hz)	δ_C_, Type	δ_H_ (*J* in Hz)
1	71.4, CH_2_	5.02, dd (12.1, 2.3)	56.4, CH_2_	4.99, d (13.7)
		4.95, d (12.1)		4.45, d (13.7)
2	125.9, C		126.1, C	
3	150.9, C		153.6, C	
4	115.3, CH	6.60, d (8.0)	114.1, CH	6.65, d (7.8)
5	130.4, CH	6.87, d (8.0)	127.7, CH	7.43, d (7.8)
6	127.0, C		121.6, CH	6.74, d (7.8)
7	142.6, C		136.7, C	
8	82.7, CH	5.39, m	131.1, CH	6.60, d (12.5)
9	38.9, CH_2_	2.04, ddd (14.8, 7.7, 2.4) 9	130.1, CH	5.93, d (12.5)
		1.78, m		
10	60.1, CH_2_	3.73, m; (7.7)	103.8, C	
		3.66, ddd (11.0, 7.7, 4.7)		
11	31.5, CH_2_	3.25, dd (15.7, 7.1)	69.3, CH	3.94, q (6.5)
		3.18, dd (15.7, 7.0)		
12	124.1, CH	5.21, t (7.0)	14.3, CH_3_	1.24, d (6.5)
13	133.3, C		67.1, CH	3.90, q (6.5)
14	25.8, CH_3_	1.73, s	14.8, CH_3_	0.98, d (6.5)
15	18.0, CH_3_	1.71, s		

**Table 2 molecules-24-02655-t002:** The ^1^H (500 MHz) and ^13^C-NMR (125 MHz) data for compounds **3** and **4**.

Position	3 ( in CD_3_OD)		4 ( in CD_3_OD)	
	δ_C_, Type	δ_H_ (*J* in Hz)	δ_C_, Type	δ_H_ (*J* in Hz)
1	65.8, CH	4.80, t (7.3)	65.3, CH	4.60, dd (5.7, 2.3)
2	123.3, C		120.7, C	
3	152.2, C		151.8, C	
4	115.5, CH	6.69, brd (8.1)	116.0, CH	6.78, brd (7.7)
5	128.2, CH	7.16, t (8.1)	128.9, CH	7.26, t (7.9)
6	122.1, CH	6.85, brd (7.9)	120.4, CH	6.89, brd (7.7)
7	137.4, C		138.9, C	
8	130.0, CH	6.67, dd (12.0, 1.7)	132.6, CH	6.90, dd (11.3, 2.0)
9	131.3, CH	6.28, dd (12.0, 3.4)	129.6, CH	6.27, dd (11.3, 4.7)
10	81.7, CH	3.93, m	78.6, CH	3.81, m
11	69.0, CH	3.94, m	68.6, CH	3.91, quint (6.4)
12	19.2, CH_3_	1.27, d (6.0)	18.3, CH_3_	1.21, d (6.4)
13 14 15	26.1, CH_3_ 97.4, C 39.4, CH_2_	1.57, s 2.32, dd (13.6, 7.0) 2.10, dd (13.6, 7.5)	26.6, CH_3_ 95.9, C 37.7, CH_2_	1.60, s 2.42, dd (14.7, 2.3) 2.17, dd (14.7, 5.7)

**Table 3 molecules-24-02655-t003:** Cytotoxic activities of compounds **1**–**7** (IC_50_ in μM).

Compound	BGC-823	Ichikawa	HepG2	7860
1	61.1	>100	20.4	>100
2	>100	>100	>100	>100
3/4	35.0	54.3	42.0	22.1
5	82.1	65.3	94.2	>100
6	78.1	58.5	85.4	>100
7	>100	>100	>100	>100
Adriamycin	1.3	1.2	1.5	2.0

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
