# Peer review of "Cytotoxic Polyketides from the Marine Sponge-Derived Fungus Pestalotiopsis heterocornis XWS03F09"

_molecules, 2019, doi:10.3390/molecules24142655_

Round 1

Reviewer 1 Report

The manuscript submitted by Lei et al describes some novel compounds isolated from a marine-derived fungus. It also describes the cytotoxicity as derived from a human cancer cell-line assay. The methods used and the conclusions drawn are all appropriate.

One question:

Line 136: What was the source of the artificial sea salt?

There are changes that are needed in the written English as indicated in Bold below, for example:

Line 20: All of the compounds…

Lines 26-27 would read better as: Marine-derived microorganisms have been attracting widespread attention as a huge resource for the discovery of bioactive natural products.

Line 30: diverse bioactivities including antitumor, antibacterial…

Line 57: oxygenated methane bonds

Line 80: compound 2 not compounds 2

Lines 81-82: I am not sure what the intent was. Maybe it should be “In addition, considering the possible biosynthesis of 2 we suggest that C-11 and C-13 are also in the S-configuration”?

Line 91: was unsuccessful

Line 95: hemiacetal derivatives

Line 97: “Material paucity” is an odd phrase, change to “lack of material”

Lines 97-98: The wording needs to be changed to something like “Due to the lack of material, the configurations of C-11 and C-15 in 3 and 4 could not be determined from single crystals and the modified Mosher’s method”

Line 99: Thus, the structures of 3 and 4 were assigned

Line 112: Compound 2 showed no cytotoxicity against the four human cancer cell lines when tested at 100 µM

Line 170: 3.6 Cytotoxicity Assay

Line 180: All of the isolated compounds

Author Response

To Reviewer #1

Q1: Line 136: What was the source of the artificial sea salt?

Reply: We appreciate valuable suggestions and comments from the reviewers. Seawater was the source of the artificial sea salt. The seawater is difficult to transported and stored. The seawater is replaced by water of the artificial sea salt.

Q2: Line 20: All of the compounds…

Reply: We have made correction according to the Reviewer’s comments in the manuscript.

Q3: Lines 26-27 would read better as: Marine-derived microorganisms have been attracting widespread attention as a huge resource for the discovery of bioactive natural products.

Reply: We agree with this point. We have revised it.

Q4: Line 30: diverse bioactivities including antitumor, antibacterial…

Reply: We have revised it.

Q5: Line 57: oxygenated methane bonds

Reply: We have revised it.

Q6: Line 80: compound 2 not compounds 2

Reply: We have revised it.

Q7: Lines 81-82: I am not sure what the intent was. Maybe it should be “In addition, considering the possible biosynthesis of 2 we suggest that C-11 and C-13 are also in the S-configuration”?

Reply: We agree with the Reviewer’s comments that " In addition, from a biosynthetic point of view, so we suggested 2 to also present the S-configuration at C-11 and C-13." should be revised to " In addition, considering the possible biosynthesis of 2 we suggest that C-11 and C-13 are also in the S-configuration ".

Q8: Line 91: was unsuccessful

Reply: We have revised it.

Q9: Line 95: hemiacetal derivatives

Reply: We have revised it.

Q10: Line 97: “Material paucity” is an odd phrase, change to “lack of material”

Reply: We agree with this point. We have revised it.

Q11: Lines 97-98: The wording needs to be changed to something like “Due to the lack of material, the configurations of C-11 and C-15 in 3 and 4 could not be determined from single crystals and the modified Mosher’s method”

Reply: We have made correction according to the Reviewer’s comments in the manuscript.

Q12: Line 99: Thus, the structures of 3 and 4 were assigned

Reply: We have revised it.

Q13: Line 112: Compound 2 showed no cytotoxicity against the four human cancer cell lines when tested at 100 µM

Reply: “Compound 2 didn’t show cytotoxicities against four human cancer cell lines at 100 mM.” was changed as “Compound 2 showed no cytotoxicity against the four human cancer cell lines when tested at 100 µM.” (Line 112).

Q14: Line 170: 3.6 Cytotoxicity Assay

Reply: “Cytotoxi” was changed as “Cytotoxicity” (Line 170).

Q15: Line 180: All of the isolated compounds

Reply: We have revised it.

Reviewer 2 Report

The review concerns the manuscript entitled: Cytotoxic Polyketides from the Marine Sponge-Derived Fungus Pestalotiopsis heterocornis XWS03F09 (molecules-546529).

My opinion is major revision.

The submitted manuscript has typical content, style, interest for readers, scientific soundness etc. The main obstacle to accept this work is lack of supplementary material containing the analysis data (NMR, MS) for the investigated compounds. I have checked the data included in the main text and in Tables 1 and 2 as well as in Figures 1-4, and they seem to be correct. However, due to lack of spectra I am not able to verify these data. Therefore, in my opinion, the work should not to be accepted for publication at this stage, unless these data will be supported.

Minor remark. Tables 2, there is no 1H NMR for H14, it should be for H15 double signal.

Author Response

Q1: The submitted manuscript has typical content, style, interest for readers, scientific soundness etc. The main obstacle to accept this work is lack of supplementary material containing the analysis data (NMR, MS) for the investigated compounds. I have checked the data included in the main text and in Tables 1 and 2 as well as in Figures 1-4, and they seem to be correct. However, due to lack of spectra I am not able to verify these data. Therefore, in my opinion, the work should not to be accepted for publication at this stage, unless these data will be supported.

Reply: We have studied reviewer’s comments carefully and have checked the  analysis data (NMR, MS) again. They are correct. Considering the Reviewer’s suggestion, the supplementary material containing the analysis data (NMR, MS) be supported.

1. Supporting Information (NMR, MS)

2. Supplementary Materials

Figure S1. The IR Spectrum of Compound 1.

Figure S2. The IR Spectrum of Compound 2.

Figure S3. The IR Spectrum of Compound 3/4.

Figure S4. The CD Spectrum of Compound 1

Figure S5. The CD Spectrum of Compound 2

Round 2

Reviewer 2 Report

The supplementary materials are included in the revised version. These materials contain the spectra of studied compounds. My main remark was fulfilled. Therefore, I recommend the manuscript for publication.